# Fatty Acid Unsaturation Degree of Plasma Exosomes in Colorectal Cancer Patients: A Promising Biomarker

**DOI:** 10.3390/ijms22105060

**Published:** 2021-05-11

**Authors:** Joan Bestard-Escalas, Rebeca Reigada, José Reyes, Paloma de la Torre, Gerhard Liebisch, Gwendolyn Barceló-Coblijn

**Affiliations:** 1Health Research Institute of the Balearic Islands (IdISBa), 07120 Palma, Spain; juan.bestard@uclouvain.be (J.B.-E.); rebeca.reigada@ssib.es (R.R.); jose.reyes@hcin.es (J.R.); paloma.delatorre@ssib.es (P.d.l.T.); 2Research Unit, University Hospital Son Espases, 07120 Palma, Spain; 3Gastroenterology Unit, Hospital Comarcal de Inca, 07300 Inca, Spain; 4Gastroenterology Department, University Hospital Son Espases, 07120 Palma, Spain; 5Institute of Clinical Chemistry and Laboratory Medicine, University Hospital Regensburg, 93042 Regensburg, Germany; Gerhard.Liebisch@klinik.uni-regensburg.de

**Keywords:** exosomes, lipidome, colorectal cancer, monounsaturated fatty acids, polyunsaturated fatty acids

## Abstract

Even though colorectal cancer (CRC) is one of the most preventable cancers, it is currently one of the deadliest. Worryingly, incidence in people <50 years has increased unexpectedly, and for unknown causes, despite the successful implementation of screening programs in the population aged >50 years. Thus, there is a need to improve early diagnosis detection strategies by identifying more precise biomarkers. In this scenario, the analysis of exosomes is given considerable attention. Previously, we demonstrated the exosome lipidome was able to classify CRC cell lines according to their malignancy. Herein, we investigated the use of the lipidome of plasma extracellular vesicles as a potential source of non-invasive biomarkers for CRC. A plasma exosome-enriched fraction was analyzed from patients undergoing colonoscopic procedure. Patients were divided into a healthy group and four pathological groups (patients with hyperplastic polyps; adenomatous polyps; invasive neoplasia (CRC patients); or hereditary non-polyposis CRC. The results showed a shift from 34:1- to 38:4-containing species in the pathological groups. We demonstrate that the ratio Σ34:1-containing species/Σ38:4-containing species has the potential to discriminate between healthy and pathological patients. Altogether, the results reinforce the utility of plasma exosome lipid fingerprint to provide new non-invasive biomarkers in a clinical context.

## 1. Introduction

Colorectal cancer (CRC) as a whole is the third-most common malignant cancer and is the fourth main cause of cancer death worldwide [1]. Even though CRC is one of the most preventable cancers, it is currently one of the deadliest. Furthermore, although the implementation of screening programs has led to a significant reduction in CRC incidence in the population aged >50 years in high-income countries, the incidence in people <50 years has unexpectedly increased significantly for causes as yet unknown [2]. Thus, it is imperative to improve early diagnosis detection strategies, as well as to monitor CRC treatment by identifying new and more precise biomarkers. While most biomarkers are based on DNA, RNA, or protein molecules, the advances in massive lipidomic analysis and the evidence that lipids can indeed provide new and more precise biomarkers are prompting a number of studies focused on these metabolites. Thus, differential lipidomic profiles have been identified by analyzing plasma from ovarian patients [3,4], prostate cancer [5,6], and CRC [7,8], or by analyzing biopsies from breast cancer [9,10] and CRC [11,12,13]. However, the application of lipid biomarkers as routine biomarkers still seems far from becoming a reality. Thus, lipid analytical techniques have been advancing rather slowly compared with the rest of “omic” techniques (e.g., transcriptomic, genomic …) There has been a substantial delay in the development of analytical techniques powerful enough to cope with lipid complexity for a long time. The systematic application of mass spectrometry (MS) methods to lipid analysis has changed this scenario, and currently, there is a large international effort to standardize aspects associated with lipid analysis, from nomenclature to analytical procedures [14]. In this already complex scenario, the recent irruption of imaging mass spectrometry techniques has added a new level of difficulty as they have unveiled that the tissue lipid profile is even more specific than was initially envisioned [15,16]. 

A second challenge is the scarce existing knowledge on lipid metabolism and the implication in pathological situations, which hampers the development of this field at the translational level. Even though at least one gene has been attributed to each of the lipid metabolic pathways, there is a shortage of information as to how any of their isoenzymes work, their specificity, or how any of these enzymes are regulated at both the activity and transcriptional levels [15,17]. Despite these difficulties, there are some successful examples of detection of altered lipid metabolism at the clinical level as the diagnosis of severe metabolic diseases in newborns by detecting aberrant levels of very long polyunsaturated fatty acids, or malnourishment by assessing n-3 fatty acids in plasma.

Among the different approaches to obtain new and complementary non-invasive biomarkers, the analysis of extracellular vesicles (EV) is experiencing considerable attention. Cells shed different types of vesicles that can be classified according to their origin: membrane vesicles and apoptotic bodies, which bleb of the plasma membrane, or exosomes, which are secreted from internal membranes through the endosomal system [18,19]. A comprehensive characterization of EVs revealed that their content is highly regulated, and originates mainly from the plasma membrane, the endocytic pathway, and the cytosol of the parental cell [20,21,22,23,24,25,26]. Interestingly, the number of circulating EV is altered in a diversity of oncologic contexts: found to be increased in ovarian and pancreatic cancer [27,28], but decreased in urine for prostate cancer patients compared with healthy controls [29]. Further, the composition of EVs is sensitive enough to identify different tumors [27,29,30] and as a treatment monitoring tool [31,32]. In terms of lipid metabolism, lipid-related enzymatic activities have been described in these vesicles, revealing a complex intercellular lipid metabolism with EVs as central players [33,34]. In particular, the authors described the presence of enzymes involved in arachidonic acid metabolism, including phospholipases A2 (cytosolic and secreted) [33,34], phospholipases C and D [33], cyclooxygenase 1 and 2 (COX1 and COX2), and lipooxygenase 12 and -15 (LOX-12 and LOX-15) [34]. Despite the fact that several studies describe the EV lipidome [23,24,25,33,35,36,37,38], only a few explore the use of the EV lipidomic signature as a source of biomarkers [6,39,40,41,42].

In previous studies, we characterized the membrane lipidome of five commercial colon cell lines and their extracellular vesicles, demonstrating that both cell and EV lipidome was able to classify cells according to their malignancy [43]. These results prompted us to investigate the use of the lipidome of EVs derived from biological fluids as a source of non-invasive biomarkers for cancer. In particular, we examined whether the lipid fingerprint could be used to assess the presence of tumors in the colon, especially in the early stages. Therefore, previously established and validated quantitative lipidomic methods using 2-phase extraction and flow-injection tandem mass spectrometry [44,45] were used to analyze plasma exosome-enriched fraction obtained from patients undergoing a colonoscopy procedure. The lipidomic results showed an evident shift from 34:1- to 38:4-containing species in the pathological groups. These changes were so relevant that the ratio of Σ34:1-containing species to Σ38:4-containing species allowed discrimination between healthy patients and patients who had already developed some kind of malignant colon lesion or were prone to do so, such as Lynch syndrome patients. Altogether, the results confirm the potential that both plasma exosomes, as a matrix of analysis, and lipidomic profiles, as the type of data generated, must provide new non-invasive biomarkers to be used in a clinical context.

## 2. Results

### 2.1. Isolated EV Membrane Lipidome Shows Changes in Phosphatidylcholine Content in Compromised Patients

The analysis of commercial cell line-derived EVs showed that the lipid composition of these vesicles is very sensitive to their origin [43]. Thus, we sought to establish whether this discriminatory potential would persist in a more complex biological matrix and context: plasma and patients. Plasma samples were collected from a cohort of patients undergoing colonoscopy and divided into the following groups: (1) Healthy group, including patients with no clinical discoveries during the colonoscopy; (2) Patients with hyperplastic polyps (HP); (3) patients with adenomatous polyps (AD); (4) Patients with invasive neoplasia (Neo, CRC patients); and (5) Patients diagnosed with hereditary colorectal cancer (Her), with no clinically relevant findings during the colonoscopy. Appendix A shows relevant clinical characteristics of the patients participating in the study. A protocol based on differential centrifugation and filtration processes was used to obtain an EV fraction highly-enriched in exosomes [46]. Although this fraction tested positive for exosome markers (CD9 antigen, 25 kDa), for simplicity it will be referred to as EVs.

To assess the differences between healthy and colon-compromised patients, we compared the membrane lipidome obtained from healthy patients and four patient groups that present a certain degree of pathologic scenario involving the colon, i.e., HP, AD, Neo, and Her. We first investigated the differences in membrane lipid classes (Figure 1). Consistent with the literature, the most abundant membrane lipids in EVs were phosphatidylcholine (PC, 36.7–53.0%, (lower and higher value)) and sphingomyelin (SM, 23.2–33.5%) (Figure 1, Appendix A) [40,42,47]. The most consistent difference between healthy and pathological groups was the increase in PC (36.7% in healthy patients vs. 46.9–53.0%. However, this difference reached significance only in AD and Her patients. Conversely, SM showed there was a clear tendency to decrease, although this difference was significant only in the Her group. In addition, we observed a clear tendency in phosphatidylinositol (PI) decreasing in AD, Neo, and Her patients compared to healthy ones (14.9%, 10.5%, and 11.8%, respectively). These results are in line with the literature [48] and in agreement with our previous study in CRC cell lines [43].

### 2.2. Plasma Isolated EV Membrane Lipid Species Composition: Impact on Polyunsaturated Fatty Acid (PUFA) Content

A comprehensive analysis at the lipid species level revealed a more complex scenario (Figure 2). The PC lipid species profile showed that the most abundant species were 34:1 (26.0–32.5%), 34:2 (14.0–23.8%), and 36:2 (9.2–13.3%) (Figure 2a and Appendix A). The Her group showed the most statistical differences when compared to the healthy group. Interestingly, numerous species in Neo and AD groups changed in the same direction, although they did not always reach statistical significance. Thus, species containing only one MUFA, like as 34:1, 36:1, and 38:1, showed decreased levels in the pathological study groups, whereas species containing di- or poly-unsaturations, such as 34:2, 36:4, 36:3, 36:2, and 38:4, were increased. The most consistent decrease was observed in 34:1 (32.5% in Healthy vs. 26.0% in AD and Neo groups, and 20.5% in Her patients (*** *p* < 0.001)) and 36:1 (10.1% in Healthy vs. 7.1% in AD, 6.4% in Neo (* *p* < 0.05), and 4.0% in Her (*** *p* < 0.001)). Conversely, the most robust increase was observed in 34:2 (14.0% in Healthy vs. 18.2% in AD, 20.0% in Neo (* *p* < 0.05), and 23.8% in Her (*** *p* < 0.001)) and 36:2 (92% in Healthy vs. 10.6% in AD, 11.9% in Neo (* *p* < 0.05), and 13.3% in Her (** *p* < 0.01)). Lysophosphatidylcholine (LPC) is the result of PC hydrolysis of one of the esterified fatty acids, commonly at the sn-2 position. The main LPC species were 16:0 (51.0–57.1%), 18:0 (16.3–19.4%), and 18:1 (10.0–13.2%) (Figure 2d and Appendix A).

The lipidomic analytical technique used does not allow to be unambiguously assign the fatty acids that are part of PC di- and poly-unsaturated species. Thus, PC 34:2 may encompass a variety of molecular species as PC 16:0/18:2 or PC 16:1/18:1, but also PC 18:2/16:0 or PC 18:1/16:1. Despite of this, in most of tissues, the *sn*-1 fatty acid tends to contain a saturated fatty acid or MUFA, while the *sn*-2 fatty acid is more often occupied by a MUFA or PUFA [49]. Taking this into account, one potential explanation of our results could be that MUFA are being substituted by a PUFA, such as 18:2n-6, 20:3n-6, or 20:4n-6, in the pathologic study group. Interestingly, these fatty acids are all enzymatically related and have been associated to the presence of malignant or premalignant lesions [11,12,50,51]. The steady LPC species levels observed throughout the groups would support this hypothesis.

The main PE species were 34:1 (7.4–23.7%), 38:4 (5.5–17.9%), and 42:7 (5.1–10.2%) (Figure 2b and Appendix A). The most consistent change was the increase in 38:4 between healthy (5.5%) and AD (13.5%), Neo (14.5%, * *p* < 0.05), and Her (17.9%, ** *p* < 0.01) patients. While not significant, there was also a slight increase in 38:5 (2.7% in healthy, 4.5% in AD, 4.2% in Neo, and 5.0% in Her) and a decrease in 38:2 (4.4% in healthy, 3.2 in AD and Her, and 3.3% in Neo), 38:1 (3.0% in healthy, 2.2% in AD and Neo, and 2.4% in Her), and 42:7 (7.8% in healthy, 5.1% in AD, 5.7% in Neo, and 7.7% in Her) species.

PI, the third most abundant lipid class (mean value of all groups 12.7%), was the lipid class showing the most robust alterations. PI was mainly composed of 38:4 (16.1–33.6%), 36:2 (15.5–19.0%), and 34:1 (8.2–14.0%) species (Figure 2c and Appendix A). The most consistent change was the significant increase in 38:4 (16.1% in healthy, 33.6% in AD, 30.1% in Neo, and 32.7% in Her), but there were also a significant decrease in saturated fatty acids, 32:0 (10.9% in healthy, 5.6% in AD, 6.6% in Neo, and 5.1% in Her) and 36:0 (10.0% in healthy, 5.2% in AD, 6.6 in Neo, and 5.1% in Her) species, and in MUFA species as 34:1 (14.0% in healthy, 8.9% in AD and Neo, and 8.2% in Her), and 36:1 (12.7% in healthy, 8.3% in AD, 8.9 in Neo, and 7.5% in Her).

Regarding sphingolipids we were able to establish the lipid species profile of SM and Cer classes (Figure 2e,f, Appendix A). The most abundant species in SM were SM 34:1;O2 (37.1–40.1%), SM 42:2;O2 (11.8–12.9%), and SM 42:1;O2 (6.1–8.8%), while for Cer they were 18:1;O2/24:0 (28.4–33.9%), 24:1 (26.7–31%), and 22:0 (12.8–13.5%). Interestingly, no significant changes or solid tendencies were detected in their species distribution between the healthy and pathological situations, except for the Her group. We detected a significant decrease in SM 34:0;O2 (1.9% in healthy, 1.4% in Her), and increases in SM 40:2;O2 (3.5% in healthy, 5.0% in Her), SM 42:4;O2 (0.1% in healthy, 0.4% in Her), and SM 42:3;O2 (3.1% in healthy, 4.9% in Her).

Altogether, these results indicate that the most affected EV membrane lipid species in a pathological situation were the ones present in the phospholipid classes. Despite the numerous solid changes in certain membrane lipid classes, none fully separated healthy from the pathological groups.

### 2.3. Identification of a Potential Biomarker for Colon Malignant Pathologies Based on Lipidomic Results

Even though the comparison between study group lipid species showed interesting changes, they were not sufficient by themselves to definitively classify all patients. However, there were some solid tendencies in pathological groups compared to healthy patients: decreases in PC 34:1, PE 34:1, and PI 34:1 levels, and increases in PC 38:4, PE 38:4, and PC 38:4 levels. For this reason, we explored whether the discrimination power of the lipidomic data could be improved. By doing so, we were able to establish a ratio, in particular the ratio of all the 34:1 containing species to all the 38:4 containing species (Figure 3). This ratio was sensitive enough to significantly separate between healthy patients and AD, Neo, and Her patients. Although HP patients showed a similar ratio to AD and Neo patients, they did not reach significant differences compared to healthy patients. Furthermore, a close look at the dispersion of the values, revealed the interesting presence of a subgroup of patients presenting a rather low 34:1/38:4 ratio, suggesting that there is a group of patients that could particularly benefit from this ratio. Unfortunately, to date, we have been unable to detect specific symptoms accounting for the formation of this subset of patients.

Finally, because of the existing overlap between various groups, we calculated the sensitivity and sensibility of this ratio using a diagnostic test evaluation calculator [52] (last accessed on 29 March 2021, Appendix A). The sensitivity calculated with the current data is 54.6%, indicating that the Σ34:1 species to Σ38:4 species ratio would detect just over half of patients with a colon lesion or prone to develop it. However, the high positive predictive value obtained (96.0%) would convey a great degree of certainty that these patients were correctly detected. Conversely, the results indicate that most healthy patients would be detected, as the specificity value was high (94.4%). However, the negative predictive value indicated that only 46.0% of them would be correctly assessed. Hence, the results are rather promising, although they also indicate that studies including larger cohorts are needed to fully understand the origin of the observed differences and the reasons why this ratio is high in some patients showing a malignant lesion or that they are prone to show it.

## 3. Discussion

Biomarkers may range from visual clinical symptoms to components of different biological fluids and tissues. The discovery of reliable, easy, and economic non-invasive biomarkers for cancer with minimum inconvenience to patients and clinicians is one of the main concerns in healthcare. Although biomarkers are commonly used as a diagnostic tool, they can also be used to stratify patients according to the evolution of their disease. The stratification of patients is of special relevance in oncology because an early surgical resection can avoid further tumor development. Therefore, the development of tools able to detect tumors before becoming invasive would definitively improve overall survival as well as reduce the economic burden associated with chemotherapy treatments.

In terms of CRC, the only non-invasive screening tool broadly applied to date is the fecal occult blood test (FOBT). FOBT relies on the fact that tumor lesions have a higher likelihood of bleeding than healthy mucosa. Current FOBT detection kits based on antibodies detect from 32 to 53% of patients with advanced neoplasia [53,54,55,56] and reduce population-screened overall CRC mortality by 22% [57]. Unfortunately, FOBT yields so many false negative and positive results that the inefficiency in the process usually overtakes digestive endoscopy departments. Therefore, the development of non-invasive biomarkers and stratification tools for this disease is still a pending subject. In this context, extracellular vesicles, particularly exosomes, are becoming an interesting biological matrix to explore. In the last few years, relative progress identifying protein or genetic biomarkers in biological fluids has occurred within the field of EV [20,23,27,28,29,30,31,32,40,42,58,59,60,61]. All these studies led to a clear conclusion: despite the exact role of EVs in pathophysiology still remaining unknown, their potential use as biomarkers is undeniable. Plasma, which is one of the most accessible biological fluids and is routinely collected in hospitals, it has some challenges when it comes to EV isolation. Thus, all tissues irrigated by the circulatory system contribute to the plasma EV pool and consequently, despite some cancers appearing to increase EV shedding [27,28], the heterogeneous fraction of healthy EV may mask the presence of tumor EV, especially in the early stages of the disease.

Tumorigenesis is a complex process affecting cell membrane lipidome at different levels. The lipid profile has proven to be specific enough to unambiguously characterize different cell states such as division [62,63], differentiation [11,12], malignization [11,12,64,65,66], and cell death [64,67], making the lipidome a powerful tool to identify biomarkers for a disease. In a clinical context, the differential lipidome, due to changes in phospholipid and sphingolipid composition, was sensitive enough to discriminate between benign breast tumors and cancer biopsies [9,10] or between healthy patients and patients with CRC [7,8]. Furthermore, membrane lipids have also been proven to be a good source of plasma lipid biomarkers in oncologic contexts. Thus, plasma lipidome from ovarian cancer patients showed differences in lysophospholipids and sphingosine-1-phosphate levels compared to healthy patients [3,4]. Similarly, plasma and urine of prostate cancer patients present alterations in PE, PS, and hexosylceramides [5,6]. Hence, if each cell type shows a specific lipid profile, it is highly plausible to consider that the shed particles, such as EV, conserve part of this specificity. In fact, there is a growing interest in establishing EV lipid composition as evidence that lipids play specific roles in EV physiology [68]. Taking into account all this evidence, together with our previous results showing that the EV lipidome do retain certain features at the compositional level of the cell origin [43], we explored the feasibility of plasma-derived EV lipidome to identify clinical biomarkers in patients at different stages of CRC.

The results shown herein point to an evident shift of some specific molecular species throughout the phospholipid classes in healthy patients and patients who had already developed some type of malignant lesion in the colon or who were prone to this, as in Lynch syndrome patients. Interestingly, the shift was rather fatty acid specific. As the lesion increased in malignancy, levels of MUFA-containing species decreased, while those species which are described to contain a high content of arachidonic acid (AA, i.e., 38:4 species) increased. This observation is in agreement with numerous studies showing increased levels of AA-species in patients suffering from cancer [11,12,50,51,69]. The latter could compromise the clinical usefulness of establishing the EV lipid profile. However, it is important to emphasize that, unlike other biomolecules, lipid species rarely appear or disappear in a pathological context, particularly in complex diseases such as cancer. Instead, the distribution of species is altered to a greater or lesser degree. For this reason, we focused on obtaining and comparing lipid profiles rather than on changes in a particular lipid species. Even though this issue should be addressed in studies including larger cohorts and different cancer types, we hypothesize that it will be possible to associate a specific lipid profile with each condition.

Thus, the increase in species presumably containing high levels of AA and the decrease in MUFA-species was observed in all the phospholipids analyzed and throughout all the pathological groups, although it did not always reach statistical significance. However, the consistency of the changes led us to explore the discrimination capacity of the relative value between Σ34:1 species to Σ38:4 species. This ratio was able to significantly separate healthy patients from patients with a malignant lesion (AD, Neo), or prone to have it (Her). The discriminatory capability of the EV lipidome is rather remarkable, particularly in the adenoma condition, as these tumors do not trespass the connective tissue and are not connected directly to the bloodstream. Both AD and Neo groups show a certain dispersion in their values that could be to do with differences within the experimental groups at the molecular level (e.g., exact mutations). Nevertheless, the presence of a subgroup of patients with a very low Σ34:1/Σ38:4 species ratio suggests it could have the ability to identify a subgroup of patients sharing common yet to be defined clinical features. More research is needed to understand the factors influencing the EV lipidome that could in turn have an impact on final patient diagnosis. Likewise, the results of the Lynch syndrome patients were rather unexpected. Interestingly, this was the group showing the largest differences compared to the healthy group despite no lesions being detected during the colonoscopy. It remains to be established what the relationship between the mutations Her patients carry and the impact on the lipidome could be.

The observed differences observed in lipid composition may be due to a wide range of reasons. On the one hand, in this study, we have analyzed the lipidome of an exosome-enriched fraction isolated from plasma. Exosomes are originated from the so-called multivesicular and released from cells after fusion with the plasma membrane. During this process, membrane lipid species are segregated, and, consequently, exosome lipidome differs from that of the parental cell. Unfortunately, the molecular mechanisms accounting for the lipid species segregation and the regulatory pathways tailoring the lipidome remain unexplored [48].

On the other hand, the exosomal fraction herein analyzed is highly heterogeneous with regard to their cell origin. It is worth mentioning that this pool is contributed not only by the irrigated tissues but also by all the circulating cells, including immune cells. Hence, one potential source of AA-enriched exosomes could be the immune cells undergoing activation processes associated with the presence of malignant cells. Furthermore, the tissue malignization impairs the epithelial permeability, implying that stromal exosomes would be more accessible. As it was previously mentioned there is evidence supporting an increase in AA-species in tumor tissues [11,12,50,51,69]. However, our results on CRC membrane lipidome established by imaging mass spectrometry techniques demonstrate that the impact of tumorigenesis differs considerably between stroma and epithelium. Thus, while we detected an increase in AA-containing species in CRC colon stroma compared to the healthy stroma, the opposite was observed between CRC and healthy epithelium [11,12]. Hence, differences on their released exosomes may be expected. Altogether, considering the current state of knowledge on how exosome lipidome is regulated, it is complex to envision the impact that a multifactorial process as tumorigenesis has on each of the types of exosomes present in plasma.

Finally, this study has certain limitations worth mentioning. First, some of the alterations did not reach significance despite showing solid tendencies, due to the differences within patients of the same study group or the low number of patients for some groups. Secondly, the origin of the healthy group should be taken into account. Thus, although the healthy group consisted of patients that did not show any endoscopic finding lesion, they were prescribed a colonoscopy because of the manifestation of certain symptoms, such as unexplained diarrheas or anemia. This fact would explain the variability in the EV “healthy” group to a certain extent. Thirdly, it is worth mentioning that in this study we have obtained the average lipidome of the plasma EV. Hence, it remains to be established if the differences observed between the study groups are due to the differential lipidome of the EV secreted by the epithelial tumor cells, the tumor microenvironment, or because of the general impact that the malignant state may have on all the circulating EV.

The increasing number of studies demonstrating the high versatility of the lipid metabolism to describe pathological situations places lipidomics at a crucial moment. This study in particular sets the basis not just to identify an EV lipid profile for CRC, but proves that the lipidome could be a reliable, and relatively easy and economical way to stratify patients according to their necessity to be submitted to other diagnostic tests. This approach to diagnosing diseases could be viable as most hospitals present the clinical analysis department with mass spectrometers. Unlike what happens with MS proteins, the fragments generated in lipidome analysis are easier to analyze once the protocol is established. Some already functioning examples of the use of lipidomic information in a clinical context include the establishment of PUFA profiles to diagnose severe metabolic diseases in newborns or malnourishment in cancer patients or patients undergoing severe surgeries.

## 4. Materials and Methods

### 4.1. Ethical Issues

The sample collection for this study was specifically approved by the Ethics Research Committee of the Balearic Islands (IB 2118/13 PI). Informed consent was obtained in written form from each patient before performing each endoscopy. Human colon biopsies were obtained in the Endoscopic Room of the Hospital Universitari Son Espases (Palma, Spain).

### 4.2. Extracellular Vesicles Isolation

To obtain the plasma from patients, the blood obtained was centrifuged at 900× *g* for 30 min at 4 °C, the supernatant plasma was kept at −80 °C until use. EV isolation protocol from plasma was performed by adapting the protocol described in Crescitelli et al. [46]. Briefly, plasma was diluted at 1:1 ratio in PBS, and then centrifuged for 10 min at 300× *g* to remove remaining cells. A second centrifugation was performed at 2000× *g* for 20 min to precipitate large cellular debris and dead cells. To discard the apoptotic bodies, the supernatant was filtered by gravity through a 0.8 µm pore size filter. Then, the volume was centrifuged at 12,200× *g* for 40 min using an ultracentrifuge (Optima L-100 XP Ultracentrifuge, Beckman Coulter, Barcelona, Spain) with a 70Ti rotor and 26.9 mL tubes (Izasa, Barcelona, Spain). The supernatant was filtered by pressure through a 0.2 µm filter. The final centrifugation to precipitate EVs was done at 120,000× *g* for 70 min with the same ultracentrifuge, rotor, and tubes than the previous step. The pellet obtained was resuspended in sterile PBS. All the processes were conducted at 4 °C. Assessment of the correct size of EVs was carried out by electronic transmission microscopy and the presence of exosomes by the expression of CD40 antigen evaluated with western blot.

### 4.3. Plasma-Derived EV Lipid Composition

Chloroform and 2-propanol were purchased from Roth (Karlsruhe, Germany) and methanol from Merck (Darmstadt, Germany). All solvents were HPLC grade. Ammonium formate was purchased from Sigma-Aldrich (Taufkirchen, Germany). Lipid extraction of plasma-derived EVs was performed according to the method of Bligh and Dyer [70] in the presence of not naturally occurring lipid species as internal standards (Avanti Polar Lipids (Alabaster, AL, USA). The following lipid species were added as internal standards: PC 14:0/14:0, PC 22:0/22:0, PE 14:0/14:0, PE 20:0/20:0 (di-phytanoyl), PS 14:0/14:0, PS 20:0/20:0 (di-phytanoyl), PI 17:0/17:0, LPC 13:0, LPC 19:0, LPE 13:0, Cer d18:1/14:0, Cer 17:0, D7-FC, CE 17:0, and CE 22:0. A total of 120 µg of EV pellet resuspended in PBS was extracted. Chloroform phase was recovered by a pipetting robot (Tecan Genesis RSP 150, Tecan Trading AG, Switzerland) and vacuum dried. The residues were dissolved either in 10 mM ammonium acetate in methanol/chloroform (3:1, *v*/*v*) (for low mass resolution tandem MS) or chloroform/methanol/2-propanol (1:2:4 *v*/*v*/*v*) with 7.5 mM ammonium formate (for high resolution MS).

Lipid analysis was performed by direct flow injection analysis (FIA) using either a triple quadrupole mass spectrometer (FIA-MS/MS; QQQ triple quadrupole, Quattro Ultima, Micromass, Manchester, UK) or a hybrid quadrupole-Orbitrap mass spectrometer (FIA-FTMS; high mass resolution, Thermo Fisher Scientific, Bremen, Germany). FIA-MS/MS (QQQ) was performed in positive ion mode using the analytical setup and strategy described previously [44,45]. A fragment ion of m/z 184 was used for PC, SM45, and lysophosphatidylcholine (LPC) [71]. The following neutral losses were applied: PE 141, PS 185, phosphatidylglycerol (PG) 189, and PI 277 [72]. PE plasmalogens were analyzed according to the principles described by Zemski-Berry [73]. Sphingosine based Cer and hexosylceramides (HexCer) were analyzed using a fragment ion of *m*/*z* 264 [74]. Correction of isotopic overlap of lipid species and data analysis was performed by self-programmed Excel Macros for all lipid classes according to the principles described previously [44,45]. Lipid species were annotated according to the recently published proposal for shorthand notation of lipid structures that are derived from MS [14]. For these data, glycerophospholipid species annotation was based on the assumption of even-numbered carbon chains only. SM species annotation was based on the assumption that a sphingoid base with two hydroxyl groups is present.

## Figures and Tables

**Figure 1 ijms-22-05060-f001:**
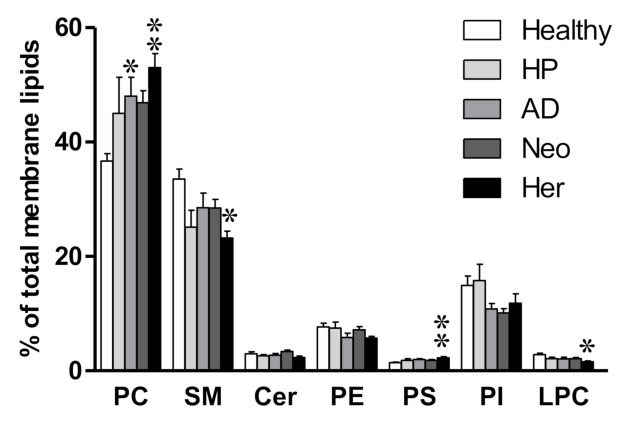
Analysis of the main membrane lipid classes of patients’ plasma derived EVs. The bars represent healthy (n = 13), with hyperplastic polyps (HP, n = 5), with adenomatous polyps (AD, n = 16), hereditary non-polyposis colorectal cancer or Lynch syndrome (Her, n = 9), and carcinoma (Neo, n = 19). Values are expressed as % of total membrane lipids (mean ± SEM). Statistical differences were assessed by one-way ANOVA followed by a Bonferroni post-test. * *p* < 0.05; ** *p* < 0.01. For simplicity, only statistical differences compared to healthy patients are represented. Detailed results showing all comparisons are included in Appendix A. Cer: ceramides, PC: phosphatidylcholine, PE: phosphatidylethanolamine, PS: phosphatidylserine, PI: phosphatidylinositol, LPC: lysophosphatidylcholine, SM: sphingomyelin.

**Figure 2 ijms-22-05060-f002:**
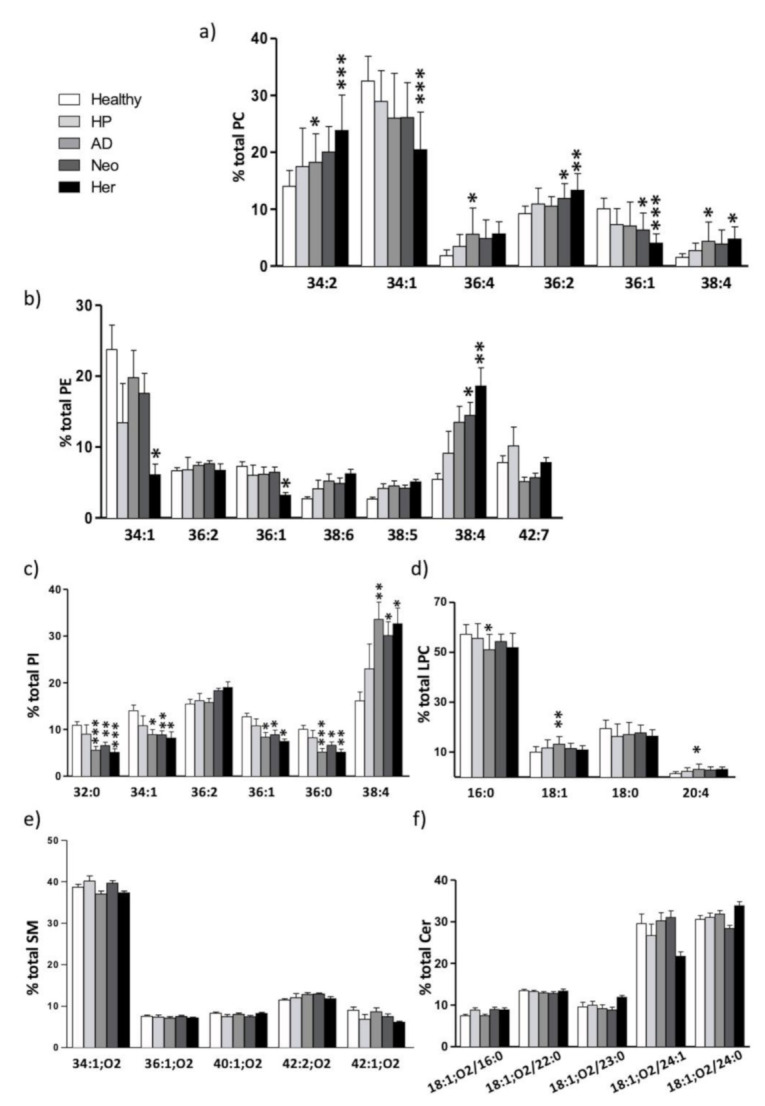
Lipid species composition of EVs isolated from patient plasma. (**a**) Phosphophatidylcholine (PC) species; (**b**) Phosphatidylethanolamine (PE) species; (**c**) Phosphatidylinositol (PI) species; (**d**) LysoPC (LPC) species; (**e**) Sphingomyelin (SM) species; (**f**) Ceramide (Cer) species. For simplicity only the species accounting for more than 5% of total species and the major AA-containing species were included. The bars represent healthy patients (n = 13), patients with hyperplastic polyps (HP, n = 5), patients with adenomatous polyps (AD, n = 16), patients of hereditary non-polyposis colorectal cancer (Her, n = 9), and carcinoma patients (Neo, n = 19). Values are expressed as % of total phospholipid or sphingolipid class (mean ± SEM). Statistical differences were assessed by one-way ANOVA followed by a Bonferroni post-test. * *p* < 0.05; ** *p* < 0.01; *** *p* < 0.001. For simplicity, only statistical differences compared to healthy patients are represented. Detailed results showing all comparisons are included in Appendix A. Cer: ceramides, PC: phosphatidylcholine, PE: phosphatidylethanolamine, PS: phosphatidylserine, PI: phosphatidylinositol, LPC: lysophosphatidylcholine, SM: sphingomyelin.

**Figure 3 ijms-22-05060-f003:**
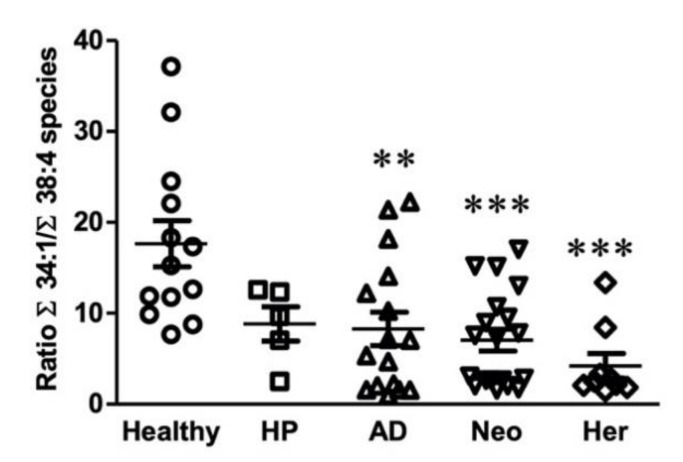
Assessment of the discriminatory capability of the ratio between the summation of 34:1 species and the summation of 38:4 species between healthy and pathological groups. The ratio was calculated by dividing the total content of 34:1 and 38:4 in PC, PE, and PI. Values are expressed mean ± SEM, n = 13 for healthy group, n = 5 for HP, n = 16 for AD, n = 9 for Her, and n = 19 for Neo. Statistical differences were assessed by one-way ANOVA followed by a Bonferroni post-test. ** *p* < 0.01; *** *p* < 0.001.

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
