# Peer review of "Fatty Acid Unsaturation Degree of Plasma Exosomes in Colorectal Cancer Patients: A Promising Biomarker"

_ijms, 2021, doi:10.3390/ijms22105060_

Round 1

Reviewer 1 Report

It seems that the Authors forgot to include the sentence.

“Therefore, previously established and validated quantitative lipidomic methods using 2-phase extraction and flow-injection tandem mass spectrometry [62,63] were used to analyze plasma
exosome-enriched fraction obtained from patients undergoing a colonoscopy procedure.”

I could not find the reference Valkone et al. (2019, https://doi.org/10.1016/j.bbalip.2019.03.011) nor the info related to: "Valkone et al. (2019, https://doi.org/10.1016/j.bbalip.2019.03.011) analyzed the lipid and protein composition of platelets and the EVs they shed. They were particularly interested in the enzymes involved in arachidonic acid metabolism and, using Western blot analysis, they were able to detect the following enzymes: cytosolic and secreted phospholipases A2 (cPLA2 and sPLA2), cyclooxygenases 1 and 2 (COX1 and COX2), and lipoxygenases 12
and 15 (12-LOX, and 15-LOX2).
We have included part of this information in the text."

Please, recheck if all the alterations mentioned in the response letter were performed in the revised manuscript.

The working hypothesis and how this and other work contributes to support it, as detailed in the Authors' answer to my last point in the previous review report, could be more clearly presented in the conclusions. 

Reviewer 2 Report

The lipid alterations associated to colorectal cancer is a very interesting issue. The authors studied the lipids in extracellular vesicles in patients with few colon disorders by mass spectrometry. The study provides some new data on lipid alterations in CRC, however the conclusions are not fully  supported by results and there is no discussion of a possible mechanism for these changes. 

Below are my comments:

  1. It is not clear if patients from this group have been diagnosed with CRC or not. There are no data on the tumour location or stage in supplementary data
  2. The list of patients groups is repeated at the and of Introduction and at the beginning of Results. Delete it in Intro or Results
  3. The Figures should be magnified
  4. Authors should clearly state what results are statistically significant - e.g. PC 34:1 is significantly lower only in EV from subjects from Her group.
  5. Figure 2 - add letters a-f to each panel
  6. Line 196 - what is DUFA?
  7. line 288 - the authors do not have the evidence that AA-containing species increased - they can only speculate that most of 38:4 species contains AA .
  8. Suppl table 1 - NOS- not otherwise specified - it is not used in this table.
  9. The conclusions in an abstract are not supported by results. First, to check if MUFA to PUFA ratio separate between healthy and CRC patients an analysis like PCA or PLSDA should be performed. Second, it is not total MUFA or PUFA but 34:1/38:4 ratio. Third, the sensitivity of 55% for diagnostics test is rather weak, and the groups studied are small. The only conclusion that is supported by the results is that EV from CRC patients have decreased 34:1/38:4 ratio. The authors should discuss the potential reasons of this fenomenon - is it a result of changed metabolism in whole patients organism or maybe the result of release of EV from tumour tissue which contains elevated PUFA levels. DOI: 10.1038/s41598-020-58895-7.

Round 2

Reviewer 2 Report

The paper has been improved according to my suggestions. For the future, I suggest giving the line numbers where the changes were made, which will facilitate the work of the reviewers.

This manuscript is a resubmission of an earlier submission. The following is a list of the peer review reports and author responses from that submission.

Round 1

Reviewer 1 Report

The work by Bestard-Escalas et al. describes the differences in the lipidome of plasma extracellular vesicles (EVs) found between healthy individuals and different groups related to colorectal cancer (CRC) at different stages, namely, with hyperplastic polyps (HP) - benign, with adenomatous polyps (AD) - premalignant, with hereditary non-polyposis colorectal cancer or Lynch syndrome (Her) and carcinoma, i.e., invasive neplasia (Neo).

Based on the alterations observed, the authors propose a MUFA/PUFA ratio found for plasma EVs as a relevant biomarker for CRC. This method of diagnosis presents several advantages, such as being derived from plasma, which is routinely collected for clinical analysis. The ratio also seems to discriminate between some of the groups studied.

The manuscript is well organized, clearly written, the different points are developed with appropriate depth, and the choice of extensive tables for supplementary material is also a positive feature, as the readers have access to the quantitative data, but the readability and understanding of the work are not hampered by their inclusion in the main text.

The experimental methods are appropriate, as well as the statistical analysis.

This is a very important piece of work, which is relevant in the context of CRC and cancer diagnosis, highlights the importance of lipidomics to disclose novel biomarkers for deadly diseases, an approach still underrated or underexploited, as mentioned by the authors. It is important that the research community dealing with cancer can access these results, as well as practitioners that may speed up the introduction of lipidomics-based assays for cancer diagnosis. The results are important from another point of view, as they add-up to the body of knowledge concerning lipid alterations in pathophysiological situations, in this case in EVs, contributing to our general understanding of the role of lipids and lipid metabolism in pathophysiological processes.

It is now asked that the authors discuss some aspects described below, and maybe add some information to the manuscript if they feel that the clarity of their work could benefit from it.

In the Introduction the authors state that “the application of lipid biomarkers as routine biomarkers still seems far from becoming a reality”. What could be the reasons for this?

In the manuscript, it can be found (page 2, lines 60-61): “lipid-related enymatic activities have been described in these vesicles”. Can the authors be more specific by giving one or two relevant examples?

In the last paragraph of the introduction, can the authors add some comments on the lipid extraction procedure and the MS methodologies used in their work, justifying their suitability?

In section 2.2 of the results it is mentioned that “the lipidomic analytical technique used does not allow the fatty acids that are part of PC di- and poly-unsaturated species to be unambiguously assigned”. Perhaps this could be more detailed for the reader that is not specialized in lipid analysis.

The lysophospholipids that the authors found were already present in the native vesicles or can they be a result of sample manipulation?

In section 2.3. of the results it can be found that: “Although HP patients showed a similar ratio to AD and Neo patients, they did not reach significant differences compared to healthy patients.” This maybe due to the small n in HP group. Is there any particular reason for the smaller n of this group compared to the other groups?

In the Discussion the authors refer that the increase in arachidonic acid maybe a general feature transversal to several cancer conditions. Since they are analyzing plasma EVs, how specific is this biomarker for CRC? Could it be common to other types of cancers and could this jeopardize its clinical usefulness?

Author Response

Response to Reviewer 1 Comments

First of all, we would like to sincerely thank the reviewer for the time taken to review our manuscript and for all the constructive comments. Next, we will proceed to address, point by point, all the issues raised by the reviewer.

Point 1: In the Introduction the authors state that “the application of lipid biomarkers as routine biomarkers still seems far from becoming a reality”. What could be the reasons for this?

Response 1: In our opinion, there are a diversity of factors. First of all, lipid analytical techniques have been advancing rather slowly compared with the rest of "omic" techniques (e.g., transcriptomic, genomic…) There has been a substantial delay in the development of analytical techniques powerful enough to cope with lipid complexity for a long time. The systematic application of mass spectrometry (MS) methods to lipid analysis has changed this scenario, and currently, there is a large international effort to standardize aspects associated with lipid analysis, from nomenclature to analytical procedures. In this already complex scenario, the recent irruption of imaging mass spectrometry techniques has added a new level of difficulty as they have clearly unveiled that the tissue lipid profile is even more specific than it was initially envisioned.

            A second challenge is the scarce existing knowledge on lipid metabolism and the implication in pathological situations, which hampers the development of this field at the translational level. Even though at least one gene has been attributed to each of the lipid metabolic pathways, there is a shortage of information as to how any of their isoenzymes work, their specificity, or how any of these enzymes are regulated at both the activity and transcriptional levels. Despite these difficulties, there are some successful examples of detection of altered lipid metabolism at the clinical level as the diagnosis of severe metabolic diseases in newborns by detecting aberrant levels of very long polyunsaturated fatty acids, or malnourishment by assessing n-3 fatty acids in plasma.

            We have added part of this reasoning in the text.

Point 2: In the manuscript, it can be found (page 2, lines 60-61): “lipid-related enymatic activities have been described in these vesicles”. Can the authors be more specific by giving one or two relevant examples?

Response 2: Valkone et al. (2019, https://doi.org/10.1016/j.bbalip.2019.03.011) analyzed the lipid and protein composition of platelets and the EVs they shed. They were particularly interested in the enzymes involved in arachidonic acid metabolism and, using Western blot analysis, they were able to detect the following enzymes: cytosolic and secreted phospholipases A2 (cPLA2 and sPLA2), cyclooxygenases 1 and 2 (COX1 and COX2), and lipoxygenases 12 and 15 (12-LOX, and 15-LOX2).

We have included part of this information in the text.

Point 3: In the last paragraph of the introduction, can the authors add some comments on the lipid extraction procedure and the MS methodologies used in their work, justifying their suitability?

Response 3: We have included the following sentence:

“Therefore, previously established and validated quantitative lipidomic methods using 2-phase extraction and flow-injection tandem mass spectrometry [62,63] were used to analyze plasma exosome-enriched fraction obtained from patients undergoing a colonoscopy procedure.

The methods were established by Liebisch et al., one of the co-authors, and were published in 2004 and 2006:

  • Liebisch, G.; Lieser, B.; Rathenberg, J.; Drobnik, W.; Schmitz, G. High-throughput quantification of phosphatidylcholine and sphingomyelin by electrospray ionization tandem mass spectrometry coupled with isotope correction algorithm. Biophys. Acta - Mol. Cell Biol. Lipids 2004.
  • Liebisch, G.; Binder, M.; Schifferer, R.; Langmann, T.; Schulz, B.; Schmitz, G. High throughput quantification of cholesterol and cholesteryl ester by electrospray ionization tandem mass spectrometry (ESI-MS/MS). Biophys. Acta - Mol. Cell Biol. Lipids 2006.

Point 4: In section 2.2 of the results it is mentioned that “the lipidomic analytical technique used does not allow the fatty acids that are part of PC di- and poly-unsaturated species to be unambiguously assigned”. Perhaps this could be more detailed for the reader that is not specialized in lipid analysis.

Response 4: We have rephrased the sentence and added some additional explanations.

Point 5: The lysophospholipids that the authors found were already present in the native vesicles or can they be a result of sample manipulation?

Response 5: Although most of the analytical procedures were carried out placing the samples in ice, or at 4ºC the lipolytic activity could not be fully excluded during preparation of vesicles. Once the exosomes were isolated, they were frozen as soon as possible at -80ºC and directly subjected to lipid extraction to minimize such potential interference.

Point 6: In section 2.3. of the results it can be found that: “Although HP patients showed a similar ratio to AD and Neo patients, they did not reach significant differences compared to healthy patients.” This maybe due to the small n in HP group. Is there any particular reason for the smaller n of this group compared to the other groups?

Response 6: The main reason is because we could not prolong the recruitment period. We agree with the reviewer that the number is rather small and different compared to the rest of the groups. However, because of the changes observed, we considered that it could be interesting to the readers as HP patients are rarely included in the studies because, from a clinical point of view, they are not considered very relevant.  

Point 7: In the Discussion the authors refer that the increase in arachidonic acid maybe a general feature transversal to several cancer conditions. Since they are analyzing plasma EVs, how specific is this biomarker for CRC? Could it be common to other types of cancers and could this jeopardize its clinical usefulness?

Response 7: This is a very interesting point that certainly needs to be addressed in future and more complex studies, including larger cohorts and different cancer types. In this scenario, we hypothesize that it will be possible to associate a stronger and more specific lipid profile with each condition.

Unlike other biomolecules, lipid species rarely “appear” and “disappear” in a pathological context, particularly in complex diseases as cancer. Instead, the distribution of species is altered, and it is for this reason that our work is focused on obtaining and comparing lipid profiles rather than on changes in a particular lipid species. In our opinion and experience, changes in arachidonic acid-containing species are always a signal that “something goes wrong”. However, the complete lipid profile is the information that would specifically indicate “what is exactly going wrong”.

This working hypothesis is based on our experiences analyzing human tissues by imaging mass spectrometry techniques, which demonstrate without a doubt that the cell lipidome is extremely unique and very sensitive to any physiological alteration. We have published this information in colon and CRC, but we also have unpublished data in circulating immune cells in CRC patients, in the context of inflammatory bowel disease, brain glioma, and our partners in melanoma skin cancer, kidney disease, spinal cord injury. Hence, if each tissue shows a specific lipid profile, it is highly plausible to consider that the particles segregated do conserve it, at least part of this specificity. Identifying these changes in a complex matrix as plasma is an additional challenge because it contains EV originated in many different cell types, and more work will be needed prior this type of approach can be used at the clinical level. However, we consider that the results generated in this study support our working hypothesis and conveys to the readers that it is worth studying this aspect.

Reviewer 2 Report

This is a relatively small lipidomic study (62 pts)  on circulating EVs from patients with various colorectal proliferative lesions. Unfortunately the clinical data are hidden in supplementary Table 1. The goal seems to me too ambitious: to find specific lipid alterations characteristic for malignant lesions... The preliminary data published have been derived form normal and cancer cell lines...and now the authors jumped directly to patient plasma. My major concern is that there is no indication, proof that the circulating EVs are epithelial cell or tumor cell derived ones...since no tumor tissue analysis was performed. It is quite irrelevant at this point to have a separate patient group of Lynch syndrome cancers...The observed differences in individual lipid species are somewhat statistically significant (not visible due to very small figure inserts), the biological relevances cannot be judged. No specificity analysis was performed for any of the described changes...nor evaluation of any of those changes for negative of positive predictive value (PPV/NPV) to discriminate the malignant lesions from benign.  If one look at the Fig3. there are large overlaps between various patient groups...in laboratory diagnostics those overlaps mean a lot of miss classifications...

Fig.2 is not labelled for A-F so any reference cannot be followed in text. In Results section  the authors mention normal cell line comparison but in MM there is no mentioning any normal cell line...

Author Response

Response to Reviewer 2 Comments

We would like to thank the reviewer for the time taken to review our manuscript. Next, we will proceed to address the issues raised by the reviewer.

Point 1: Unfortunately the clinical data are hidden in supplementary Table 1.

Response 1: Clinical data were placed in a supplementary Table as it was done in the previous publications following the suggestions of the reviewers.

Point 2: The goal seems to me too ambitious: to find specific lipid alterations characteristic for malignant lesions

Response 2: First, we consider that it is worth acknowledging that the idea to identify lipid biomarkers for disease in EVs is not original from our research group and some publications are already available. Thus, there are a number of studies pursuing this aim in a variety of study models (cancer cell lines, doi: 10.1186/s13048-020-0609-y; doi: 10.1016/j.bbagrm.2012.08.016.) and biological fluids (urine, doi: 10.1016/j.ejca.2016.10.011;  plasma, doi: 10.1038/s41598-020-73411-7 ). The number of articles aiming to identify protein or mRNAs biomarkers in plasma EV is considerably higher, which would be consistent with the historical delay in the study of the membrane lipidome in a clinical context.

Our working hypothesis is based on our experience demonstrating that the cell lipidome is highly unique and very sensitive to any physiological alteration. Our rationale is that if each cell type shows a specific lipid profile, it is highly plausible to consider that the particles segregated conserve at least part of this specificity.

Point 3: My major concern is that there is no indication, proof that the circulating EVs are epithelial cell or tumor cell derived ones...since no tumor tissue analysis was performed.

Response 3: This is a very interesting point that certainly needs to be addressed in the future. However, in our opinion, it does not undermine the results showed in this study and their potential. As we mentioned in the text, identifying these changes in a complex matrix as plasma is an additional challenge because it contains EV originated in many different cell types. Isolation of EV from colon biopsies is more complicated than in other tissues because of the high microbial content included in the biopsies. In addition, colon biopsies contain different cell types (epithelial, fibroblasts, infiltrated immune cells), each of their specific lipidome, and shedding most probably EV with their own lipidome. Unfortunately, the option to cultivate healthy and tumor colon organoids (which would contain only epithelial cells) has a clear disadvantage because the culturing conditions alter both cells and EV lipidome. We are exploring diverse options to circumvent these drawbacks.

Point 4: The observed differences in individual lipid species are somewhat statistically significant (not visible due to very small figure inserts), the biological relevances cannot be judged.

Response 4: We have increased the size of the figure so that the inserts become more visible.

Point 5: No specificity analysis was performed for any of the described changes...nor evaluation of any of those changes for negative of positive predictive value (PPV/NPV) to discriminate the malignant lesions from benign.  If one look at the Fig3. there are large overlaps between various patient groups... in laboratory diagnostics those overlaps mean a lot of miss classifications...

Response 5: We have acknowledged in the manuscript that the studies have a series of limitations including the overlap between various groups of patients. There is no doubt that before incorporating the EV lipidome as a diagnostic tool, more studies, including larger cohorts should be carried out. However, in our opinion, the results showed herein support our working hypothesis and convey to the readers that it is worth studying the EV lipidome as a source of biomarkers.

Point 6: Fig.2 is not labelled for A-F so any reference cannot be followed in text. In Results section  the authors mention normal cell line comparison but in MM there is no mentioning any normal cell line...

Response 6: We have corrected this mistake.

Round 2

Reviewer 2 Report

The two major points raised have not been corrected:

  1. The source of EV in the circulation: tumor, stroma or other cell derived is not known no comparison to tumor tissue
  2. Positive or negative predictive value of any lipid species to discriminate normal from tumor is not done..